# Exploring Mechanisms of Antifungal Lipopeptide Iturin A from Bacillus against *Aspergillus niger*

**DOI:** 10.3390/jof10030172

**Published:** 2024-02-24

**Authors:** Shiyi Wang, Min Xu, Ye Han, Zhijiang Zhou

**Affiliations:** School of Chemical Engineering and Technology, Tianjin University, Tianjin 300350, China; wangshiyi1010@126.com (S.W.); minxu@tju.edu.cn (M.X.); hanye@tju.edu.cn (Y.H.)

**Keywords:** iturin A, *Aspergillus niger*, inhibitory mechanism, proteomic

## Abstract

The control of *Aspergillus niger* (*A. niger*) is of great significance for the agricultural economy and food safety. In this study, the antifungal effect and mechanism of iturin A from *Bacillus amyloliquefaciens* (CGMCC No. 8473) against *A. niger* (ATCC 16404) were investigated using biochemical analyses and proteomics. Changes in a mycelium treated with iturin A were observed using scanning electron microscopy and transmission electron microscopy, including mycelial twisting and collapse, organelle disintegration, and intracellular vacuolization. The cytomembrane integrity of *A. niger* was affected by iturin A, as detected by propidium iodide staining. In addition, the generation of excess reactive oxygen species, the hyperpolarization of the mitochondrial membrane potential and malondialdehyde accumulation also indicated that iturin A induced apoptosis in *A. niger* through the oxidative stress pathway. Proteomics results showed that 310 proteins were differentially expressed in the *A. niger* mycelium exposed to iturin A, including 159 upregulated proteins and 151 downregulated proteins, which were mainly associated with energy metabolism of *A. niger*. We propose that iturin A might inhibit the growth of *A. niger* by disrupting cytomembrane integrity, via oxidative stress, and by interfering with glycolysis/gluconeogenesis and the tricarboxylic acid cycle. Overall, iturin A is a promising antifungal agent that provides a rationale for controlling *A. niger* contamination in food.

## 1. Introduction

Fungal contamination has had a significant impact on agricultural economic development worldwide [1]. According to the Food and Agriculture Organization of the United Nations (FAO), 5% of the world’s cereals lose their food value yearly due to fungal infestation. *Aspergillus niger* (*A. niger*) is one of the most common fungi causing the deterioration of cereals, and black mold reduces the quality of agricultural products and their derivatives, resulting in serious economic losses [2]. In addition, it is crucial to note that *A. niger* poses a serious threat to human health due to a variety of pathogenic and carcinogenic toxins produced during the infestation process [3].

Recently, the development of effective approaches to suppress cereal contamination has become an active field of research, with a large proportion of chemical fungicides [4]. However, the overuse of chemical fungicides leads to the development of fungal resistance and the accumulation of toxic chemical residues in human bodies and ecosystems owing to the difficulty of degrading in the environment [5]. As a consequence, biocontrol agents (BCAs), including microorganisms and their secondary metabolites, have attracted the attention of researchers for their environmental friendliness, safety, and high efficiency.

*Bacillus* sp. is considered an effective and promising BCA due to its secretion of antifungal metabolites [6]. The antimicrobial peptides of *Bacillus* sp. mainly consist of iturins, surfactins and fengycins, exhibiting broad-spectrum antimicrobial activity, low toxicity, and hypoallergenicity [7]. All three molecules are synthesized by non-ribosomal means [8]. The iturin family comprises iturins A, C, D, and E, bacillomycins D, F, and L, bacillopeptin, and mycosubtilin, which consists of a cyclic heptapeptide and an α-amino fatty acid chain with 14–17 carbon atoms [9]. Iturin A has been reported to have strong antimicrobial activity against a variety of fungi, such as *Penicillium digitatum* [10], *Colletotrichum gloeosporioides* [11], and *Botrytis cinerea* [12]. Moreover, *Bacillus amyloliquefaciens* (*B. amyloliquefaciens*) BH072 (isolated from honey in our previous study) secretes an antifungal lipopeptide which has a strong inhibitory effect on *A. niger* [13]. However, to the best of our knowledge, there are currently no studies in the literature reporting the mechanism of inhibiting *A. niger* of iturin A, limiting the effective utilization of iturin A in the food field.

The objective of the present study is to reveal the inhibitory mechanism of iturin A treatment against *A. niger*. We present the effects of iturin A on biochemical factors including cytomembrane integrity, reactive oxygen species (ROS) accumulation, mitochondrial dysfunction, and mitochondrial enzyme activity. Differential protein expression and metabolic regulation were investigated by a proteomic analysis. *A. niger* was treated with iturin A at its minimum inhibitory concentration (MIC), which verified the biochemical changes in the mycelium exposed to iturin A. This study provides novel insights into the antifungal mechanisms of iturin A on *A. niger* and promotes its utilization as a promising antifungal agent in controlling *A. niger* contamination.

## 2. Materials and Methods

### 2.1. Chemicals

Propidium iodide (PI) was purchased from Macklin (Shanghai, China). DCFH 2′,7′-dichlorodihy drofluorescein diacetate (DCFH-DA) was purchased from Beyotime (Shanghai, China). Rhodamine 123 (Rh123), Luria–Bertani (LB) medium, potato dextrose agar (PDA), and potato dextrose broth (PDB) were obtained from Solarbio (Beijing, China). Other chemicals were obtained from the Sigma-Aldrich Company (St. Louis, MO, USA).

### 2.2. Microorganisms and Culture Conditions

*B. amyloliquefaciens* BH072 was previously isolated and identified from honey [13]. The strain was deposited in the Microbiology Laboratory of School of Chemical Engineering, Tianjin University, and was cultured in LB medium at 37 °C.

*A. niger* ATCC16404 was obtained from the China Industrial Microbial Strain Collection Management Center (CICC). The strain was inoculated into PDA medium and incubated in a constant-temperature incubator at 28 °C for 48 h. After maturation, the spores were washed off with sterile water and diluted to a spore suspension of 10^6^ spores/mL by a hemocytometer [14].

### 2.3. Extraction of Iturin A

The fermentation of *B. amyloliquefaciens* BH072 was performed in 500 mL conical flasks with 300 mL of LB medium. The cultures were incubated in a rotary incubator shaker for 48 h at 30 °C and at 160 rpm. After incubation, the precipitate was removed by centrifugation (4 °C, 4000 rpm, and 20 min), and the pH of the cell-free supernatant of the fermentation broth was adjusted to 2.0 with 6 M HCl, with slight stirring or standing overnight. The precipitate was then recovered by centrifugation (4 °C, 4000 rpm, and 20 min) and extracted with methanol, and the extract solution was evaporated to dryness using a rotary evaporator and dissolved with PBS (10 mM, pH 7.4) to obtain a crude extract. The extract was purified and identified using Sephadex LH-20 column chromatography (Yuanye, Shanghai, China) and a reversed-phase high-performance liquid chromatography (RP-HPLC) instrument equipped with a C18 solid-phase extraction column (Venusil XBP, 5 μm, 2.1 mm × 150 mm, Agela Corporate, Torrance, CA, USA). In this study, iturin A was extracted according to the previously described method [13], and the extract was finally passed through a filter (pore size: 0.22 μm) and stored at −20 °C.

### 2.4. Determination of the Inhibitory Effect of Iturin A on A. niger

#### 2.4.1. Determination of Minimum Inhibitory Concentrations

The minimum inhibitory concentration (MIC) was determined using the twofold dilution method with a slight modification [15]. Iturin A was diluted to different concentrations with PDB medium. A mixture formed using 100 μL of spore suspension (10^6^ spores/mL) and 100 μL of iturin A was then inoculated to a 96-well plate. The negative control was the addition of only 200 μL of PDB medium, and the positive control was 100 μL of spore suspension (10^6^ spores/mL) and 100 μL of PDB medium. The 96-well plates were incubated in an incubator at 28 °C for 24 h. The lowest inhibitory concentration at which no visible colonies appeared to the naked eye was defined as the MIC.

#### 2.4.2. Effect of Iturin A on Spore Germination

The inhibition of spore germination was conducted according to previously published methods, with slight modifications [16]. In a nutshell, 100 μL of spore suspension (10^6^ spores/mL) was inoculated to 100 μL of PDB medium containing iturin A at MIC values of 0, 1/2MIC, MIC, and 2MIC. To ensure sufficient moisture, the concave slide was placed into a Petri dish lined with two layers of sterile wet filter paper. The mixture was poured into the center of the Petri dish and incubated for 10 h at 28 °C. At least 200 spores per group were observed under a light microscope, and the number of spores germinating was recorded.

#### 2.4.3. Effect of Iturin A on Colony Growth

The inhibitory effect of iturin A on mycelial growth was detected using a previously reported method [17]. A sample of PDA medium containing different concentrations of iturin A (0, 1/2MIC, MIC, and 2MIC) was poured into Petri dishes (60 mm in diameter), 10 μL of *A. niger* spore suspension (10^6^ spores/mL) was inoculated in the center of the medium, and the plates were incubated at 28 °C. Photographs were taken, and vernier calipers were used to measure the colony diameter of each group as the control mycelium grew to the edge of the Petri dish.

### 2.5. Scanning Electron Microscopy (SEM) and Transmission Electron Microscopy (TEM) Observations

The spore suspension (10^6^ spores/mL) was inoculated in PDB medium and shaken for 48 h at 28 °C. The incubation was continued with the addition of iturin A (final concentration: MIC) for 12 h. After centrifugation (4 °C, 6000× *g*, 5 min), the mycelium was collected and washed twice with distilled water. The supernatant was discarded, and the mycelium was first immersed in 2.5% glutaraldehyde, vortexed for 30 s, and left for 12 h at 4 °C for fixation. The samples were then washed three times with PBS (10 mM, pH 7.4) for 15 min each and dehydrated in a graded ethanol series of 30%, 50%, 60%, 70%, 80%, 90%, and 100% (*v*/*v*) ethanol for 15 min each. The above-pretreated mycelium was obtained for further use.

For SEM, after graded dehydration, the ethanol in the samples was replaced with isoamyl acetate, and the samples were transferred to a vacuum freeze dryer for full drying. Finally, the powdered mycelium was adhered to the SEM carrier table with conductive adhesive and sprayed with gold, and the samples were observed under an SEM instrument (S4800, Hitachi, Chiyoda City, Japan).

TEM was used to observe the effect of iturin A on the internal structure of the mycelium. The pretreated mycelium was further treated with an acetone solution (100%) for 20 min. After embedding by sectioning, the sections were double-stained with uranyl acetate and lead citrate for 10 min. The samples were observed under a TEM instrument (H-7650, Hitachi, Chiyoda City, Japan) after drying.

### 2.6. Determination of Cytomembrane Damage

Cytomembrane integrity was visualized using propidium iodide (PI) according to the method previously described [18]. The spore suspension (10^6^ spores/mL) was inoculated into the PDB and incubated for 48 h before adding iturin A to final concentrations of 0, 1/2MIC, MIC, and 2MIC, and incubation was continued for 12 h at 28 °C with 150 rpm shaking. The mycelium was rinsed three times with PBS (10 mM, pH 7.4) and then stained with 10 μg/mL PI for 30 min in the dark, followed by observation under a fluorescence microscope (Ti2-U, Nikon, Tokyo, Japan). The excitation and emission wavelengths of the PI staining were adjusted to 493 nm and 636 nm, respectively.

### 2.7. Intracellular Reactive Oxygen Species Accumulation

2,7-Dichlorodihydrofluorescein diacetate (DCFH-DA) was used to measure the effects of iturin A on the accumulation of ROS [18]. Fungal suspensions (10^6^ spores/mL) were incubated under standard conditions, as described in Section 2.6. After iturin A treatment, the mycelium was washed three times with PBS (10 mM, pH 7.4) and incubated with 10 μM of DCFH-DA at 28 °C for 30 min in the dark. Fluorescence images of the mycelium were observed using a fluorescence microscope (Nikon Ti2-U, Tokyo, Japan). The DCFH-DA staining fluoresced at an excitation wavelength of 488 nm and an emission wavelength of 525 nm.

### 2.8. Determination of Mitochondrial Membrane Potential

The mitochondrial membrane potential in *A. niger* was detected using the fluorescent dye Rh123 and a flow cytometer according to the previously described method [19]. *A. niger* spore suspension (10^6^ spores/mL) was treated with various concentrations of iturin A (0, 1/2MIC, MIC, and 2 MIC) for 12 h. A final concentration of 15 μM of Rh 123 was added to the treated spore suspensions and incubated at 28 °C for 30 min in the dark. Furthermore, the suspensions were washed, resuspended in PBS (10 mM, pH 7.4) and analyzed by flow cytometry (FACS Aria III, BD, Franklin Lakes, NJ, USA). The maximum excitation and emission wavelengths of Rh123 were 507 nm and 529 nm, respectively.

### 2.9. Determination of Malondialdehyde (MDA) Content

The MDA content was detected by the thiobarbituric acid (TBA) method according to the previous method [20]. The spore suspension (10^6^ spores/mL) was incubated under standard conditions as described in Section 2.6, and the mycelium was collected. Next, 1 g mycelial samples treated with different concentrations of iturin A were added to the extract liquid. The extracts were homogenized in an ice bath and then centrifuged (4 °C, 8000× *g*, and 10 min). The supernatant was obtained, and the MDA content was measured according to the kit instructions (Solarbio, Beijing, China).

### 2.10. Determination of Mitochondrial ATPase and Dehydrogenase Activity

The ATPase activity and mitochondrial dehydrogenase exposure to iturin A were determined according to previously described methods [21]. As previously mentioned, mycelium exposed to iturin A was collected. A sterile solution containing 50 mM of Tris (pH 7.5), 2 mM of EDTA (ethylenediaminetetraacetic acid), and 1 mM of PMSF (phenylmethylsulfonyl fluoride) was used to suspend the mycelium. Then, the mycelium was broken using a high-speed homogenizer for 90 s and centrifuged (4 °C, 3000× *g*, and 10 min) to remove cellular debris. Finally, the supernatant was centrifuged (4 °C, 12,000× *g*, and 40 min), and the mitochondria were resuspended in a buffer at 4 °C until use. ATPase and mitochondrial dehydrogenase activity levels were determined using a malate dehydrogenase (MDH) kit (Solarbio, Beijing, China), a succinate dehydrogenase (SDH) kit (Nanjing Jiancheng Bio. Inst., Nanjing, China), and an ATPase kit (Nanjing Jiancheng Bio. Inst., Nanjing, China), respectively.

### 2.11. Proteomic Analysis

#### 2.11.1. Protein Extraction

The spore suspension (10^6^ spores/mL) was suspended in PDB medium and incubated at 28 °C for 48 h with 160 rpm shaking. Subsequently, iturin A at a final concentration equal to its MIC (25 μg/mL) was added to the medium and incubated for 12 h. Then, 1 g of mycelium was weighed and added to pre-cooled mortar, and liquid nitrogen was added to fully grind it into powder. Each group of samples was dispersed via ultrasonication by adding a 4-fold volume of extraction buffer (containing 10 mM of dithiothreitol and 1% protease inhibitor) and centrifuged (4 °C, 6000× *g*, and 10 min) to remove the supernatant, then precipitated overnight by adding a 5-fold volume of 0.1 M of ammonium acetate/methanol; the protein precipitates were then washed with methanol and acetone, respectively. The final precipitate was reconstituted with 8 M of urea, and the protein concentration was determined using a BCA kit (Solarbio, Beijing, China). Labelfree proteomics assay reference methods for proteomics assays were detailed in the Appendix A.

#### 2.11.2. Enzymatic Desalination

An appropriate amount of total protein was taken and added to a reducing agent buffer for 1 h at 37 °C. The final concentration of 50 mM of Iodoacetamide (IAA) was added, and the reaction was carried out for 45 min at room temperature in the dark. Subsequently, the samples were quenched by adding 1M of DTT solution and IAA. The final concentration of urea was made less than 1 mol/L with 50 mmol/L of an NH_4_HCO_3_ solution and incubated overnight at 37 °C with a sequencing-grade trypsin solution. Finally, the digested peptides were desalted using a C18 solid-phase extraction column.

#### 2.11.3. LC-MS/MS and Bioinformatic Analysis

Peptides were dissolved in phase A of a liquid chromatography mobile phase and separated using an EASY-nLC 1200 (Thermo Fisher Scientific, Waltham, MA, USA), using a gradient elution mode at a flow rate of 300 μL/min with eluent B (acetonitrile containing 0.1% formic acid) and eluent A (an aqueous solution containing 0.1% formic acid) for 120 min. The peptides were injected into the NSI ion source for ionization and then analyzed using a Q Exactive HF (Thermo Fisher Scientific, Waltham, MA, USA) mass spectrometer. Ultimately, the data were acquired with a selected mass range of 400–1800 *m*/*z*. The Gene Ontology (GO) of the protein was analyzed by the InterProScan-5 program. A genomic (KEGG) database was used to analyze protein families and pathways.

### 2.12. Statistical Analysis

All measurements were conducted at least three times, and data are shown as mean ± standard deviation values. Statistical analyses were performed using a one-way analysis of variance (ANOVA), using IBM SPSS Statistics Software 26.0 (IBM Corporation, Armonk, NY, USA), with *p* values < 0.05 considered statistically significant. Graphs were created using Origin 9.0 software.

## 3. Results

### 3.1. Inhibitory Effect of Iturin A on A. niger

#### 3.1.1. Effect of iturin A at the Minimum Inhibitory Concentration against *A. niger*

The minimum inhibitory concentration (MIC) is a quantitative index used to evaluate the activity of antimicrobial substances. Figure 1A shows that iturin A could effectively inhibit the growth of *A. niger* at 25 μg/mL with no colonies visible to the naked eye. As a consequence, the MIC of iturin A against *A. niger* was 25 μg/mL.

#### 3.1.2. Spore Germination and Mycelial Growth Assay

The antifungal effects of different concentrations of iturin A on spore germination and the mycelial growth of *A. niger* are shown in Figure 1. As illustrated in Figure 1B, iturin A dose-dependently decreased spore germination at all tested concentrations. After 10 h of incubation, almost all spores germinated in the control group (Figure 1B(a)), whereas only 7.6% of spores germinated in the 2MIC group (Figure 1B(d)). Similarly, mycelial growth was validly inhibited as the iturin A concentration increased (Figure 1C). The colony diameters of the *A. niger* incubated on the PDA plates covered with 1/2 MIC, MIC, and 2 MIC concentrations of iturin A were 72%, 77.16%, and 83.83% lower than that of the control, respectively. Therefore, iturin A has a significant inhibitory effect on the spore germination and mycelial growth of *A. niger* in a dose-dependent manner.

### 3.2. Effect of Iturin A on the Morphology and Cellular Structure of A. niger

To characterize the surface structure of the mycelium, SEM was used to reveal changes in the mycelium [22]. In the control group (Figure 2A), the untreated mycelium was morphologically intact, with uniform thickness and a smooth surface. However, in the MIC-treated group (Figure 2B), the mycelium showed an obviously twisted deformation and irregular folding. In addition, occurrences of mycelial drying and depression could be observed which might be due to the rupture of mycelial cytomembrane, leading to the leakage of inclusions. These results suggest that the presence of iturin A could cause irreversible changes in the morphology of a mycelium, destroying the normal structure of the mycelium and thus affecting its physiological function.

We further observed alterations in the internal ultrastructure of the mycelium using TEM. In the control group (Figure 2C), the mycelium exhibited a normal structure characterized by an intact cytomembrane and cell wall, and the organelles were evenly and orderly distributed within the cytoplasm. However, after iturin A treatment (Figure 2D), the cells were vacuolated, and a large number of organelles were accumulated, resulting in the blurring of cell boundaries and the loss of the original morphology. In particular, compared to Figure 2E showing an intact mitochondrial structure, Figure 2F displays a blurred mitochondrial structure with severe lysis. We speculated that iturin A severely affected the internal ultrastructure of the mycelium, in particular damaging the structure and function of cytomembranes and mitochondria.

### 3.3. Effect of Iturin A on Cytomembrane Integrity

PI is a red fluorescent fuel that can cross damaged cytomembranes and specifically bind double-stranded DNA, resulting in red fluorescence [23]. As shown in Figure 3A, the fluorescence was nearly undetectable in the control group (Figure 3A(a)), while fluorescence could be observed in the mycelium treated with iturin A. Moreover, the fluorescence intensity increased continuously with increasing treatment concentration (Figure 3A(b–d)). The results imply that iturin A disrupted the integrity of the cytomembrane.

### 3.4. Effect of Iturin A on Intracellular ROS Generation

ROS are intermediates of cellular oxidative metabolism and are formed by mitochondria, and the excessive production and accumulation of ROS is one of the hallmarks of apoptosis [24]. Compared to the control group, the green fluorescence intensity was dose-dependently enhanced following treatment with an increased concentration of iturin A (Figure 3B). Consequently, the results indicate that iturin A caused ROS accumulation in *A. niger*, potentially leading to fungal death.

### 3.5. Effect of Iturin A on the Mitochondrial Membrane Potential

The mitochondrial membrane potential (MtΔψ) is a sensitive indicator of cellular energy status. Rhodamine 123 (Rh123) is a permeable cationic dye that can selectively enter the mitochondrial matrix and reflect changes in the MtΔψ. As shown in Figure 4A, iturin A increased the fluorescence intensity of Rh123 in a concentration-dependent manner after 12 h of incubation. The fluorescence intensity of the spores was only 0.16% in the untreated condition (Figure 4A(a)), whereas it reached 10.5%, 23.5%, and 33.6% after 1/2, MIC, and 2 MIC iturin A treatments (Figure 4A(b–d)), respectively. These results indicate that iturin A induced the hyperpolarization of the MtΔψ, resulting in mitochondrial dysfunction.

### 3.6. Effect of Iturin A on MDA Content

MDA is a lipid peroxidation product that is commonly used to indicate the degree of lipid peroxidation [25]. Figure 4B illustrates the content of MDA in the cytomembrane of *A. niger*. Compared with the control, the MDA content increased significantly with the iturin A concentration. Notably, the MDA content in mycelia increased to 4.689 nmol/g, 6.327 nmol/g, and 9.642 nmol/g, respectively, after incubation with iturin A at 1/2MIC, MIC, and 2MIC for 12 h. It is noteworthy that the content of MDA in the 2M group was 28 times higher than that in the control group. These results clearly showed that iturin A promoted cytomembrane lipid peroxidation and stimulated MDA formation.

### 3.7. Determination of Mitochondrial Dehydrogenase and ATPase Activity

The tricarboxylic acid (TCA) cycle is the major pathway of fungal mitochondrial metabolism, and the disruption of this process can result in microbial growth suppression or even apoptosis. The catalytic enzymes succinate dehydrogenase (SDH) and malate dehydrogenase (MDH) play crucial roles in the TCA cycle, and they are usually the main targets of fungicidal activity. As shown in Figure 4C,D, after iturin A exposure, the activity levels of MDH and SDH decreased continuously with the increase in iturin A concentration. Compared with the control group, the inhibition of SDH activity at iturin A concentrations in the 1/2MIC, MIC, and 2MIC groups was 21.33 ± 4.94%, 35.04, ± 3.81% and 84.86 ± 4.57%, respectively. Similarly, The MDH activity was considerably reduced at these concentrations of iturin A, ranging from 52.49 ± 7.47% to 85.49 ± 3.24%. These findings demonstrate that *A. niger* treatment with iturin A resulted in significant decreases in SDH and MDH activity levels, thereby interfering with the TCA cycle and affecting the respiratory metabolism of the fungus.

ATPase is also a key enzyme in respiration and energy metabolism in fungi. Changes in ATPase activity in *A. niger* are shown in Figure 4E. Compared with the control group, iturin A exposure at 1/2MIC, MIC, and 2MIC significantly reduced the ATPase activity by 39 ± 2.99%, 61.59 ± 2.80%, and 93.34 ± 2.89%. This result reveals that iturin A can inhibit the activity of mitochondrial ATPase, thereby suppressing ATP synthesis and inducing apoptosis in *A. niger*.

### 3.8. Comparative Proteomic Analysis of A. niger

#### 3.8.1. Proteomic Profiles

Figure 5 shows a volcano map of *A. niger* treated with iturin A. The green dots indicate downregulated differential proteins, the red dots indicate upregulated differential proteins and the gray dots indicate non-significant differential proteins. In comparison to the control group, 310 proteins were significantly expressed in the iturin A treatment group, of which 159 proteins were upregulated and 151 proteins were downregulated.

#### 3.8.2. GO Function Enrichment of Differentially Expressed Proteins

The identified proteins were annotated and enriched by GO. GO functional annotation covers three aspects of biology: biological processes (BPs), cellular components (CCs) and molecular functions (MFs). As shown in Figure 6, the membrane accounted for the highest proportion, followed by the cytoplasm and nucleus in the CC classification. Translation and proteolysis were enriched in the BP classification. In the MF classification, ATP binding accounted for the highest proportion, followed by metal ion binding.

#### 3.8.3. KEGG Function Enrichment of Differentially Expressed Proteins

Based on the KEGG database, the metabolic pathways of the differential proteins after iturin A treatment were analyzed, and the top 10 pathways are shown in Figure 7A. The metabolic pathways were divided into two main categories. The first metabolic pathway was amino acid biosynthesis and degradation, including the biosynthesis of amino acids, valine, leucine, and isoleucine. Other types of metabolic pathways were associated with energy metabolism, mainly glycolysis/gluconeogenesis, the TCA cycle, and oxidative phosphorylation. 2-oxocarboxylic acid metabolism, carbon monoxide metabolism, glyoxylate and dicarboxylate metabolism, and pantothenate and CoA biosynthesis were also involved in the KEGG enrichment results.

Saccharides are an essential source of energy required to sustain the life activities of microorganisms. Glycolysis/gluconeogenesis is one of the major pathways through which organisms acquire energy. Therefore, we enriched and analyzed the effect of iturin A treatment on the glycolysis/gluconeogenesis pathway through the KEGG pathway. The results are illustrated in Figure 7B. In the glycolysis/gluconeogenesis pathway, eight proteins were enriched: two up-regulated proteins and six downregulated proteins.

## 4. Discussion

The application of BCAs, including bacteria and secondary metabolites, is regarded as an effective biocontrol strategy to control food fungal contamination. Iturin A, a metabolite released by the genus *Bacillus*, has an inhibitory impact on filamentous fungus and has outstanding potential for application in agricultural and food production. In this study, the antifungal mechanism of iturin A against the growth of *A. niger* was investigated using biochemical analyses and proteomics.

The fungal cytomembrane is an important barrier against the free entry of extracellular substances into the cell which ensures the exchange of substances and energy as well as maintains the viability of the cell and the orderly conduct of physiological reactions. To date, the cytomembrane is the target of most antimicrobial peptides found in nature [26]. Regarding iturin A, owing to its amphiphilic nature, it can bind to the lipid bilayer in the cytomembrane to form different secondary structures for its repressive expression [27]. Based on PI staining and electron microscopy, we reinforced that the apparent distortion and depression of the mycelium after iturin A treatment is due to the disruption of the cytomembrane, accompanied by the leakage of its contents. In proteomics, this can be confirmed by GO functional annotation results, which show that the cytomembrane of *A. niger* contains the most differential proteins among cellular fractions. Moreover, previous reports have shown that the iturin family can disrupt fungal cytomembranes, causing the leakage of contents such as nucleic acids, proteins, and K^+^ [12]. Hence, it can be concluded from the results that the disruption of cytomembrane integrity is one of the main mechanisms through which iturin A inhibits the growth of *A. niger*.

ROS-induced cellular oxidative damage is a common antifungal mechanism in *Candida* [28]. There is mounting evidence that the excessive accumulation of ROS in a mycelium also causes severe oxidative damage to the inner membrane and organelles (e.g., mitochondria) of filamentous fungal cells [29]. In DCFH staining assays, the mycelium exposed to iturin A displayed increasing fluorescence intensity, implying a gradual increase in ROS levels. Similarly, in previous studies, cinnamaldehyde was found to induce the accumulation of ROS in *A. niger* cells, ultimately leading to cytomembrane lipid peroxidation to impede the development and growth of fungal cells [14]. In addition, excess ROS also affect intracellular biomolecules, such as cytoskeletons, DNA, and proteins [30]. Therefore, we hypothesize that ROS-mediated oxidative stress may be one of the main mechanisms through which iturin A exerts its antifungal effects. To validate the mechanism of oxidative stress, we further investigated the characteristic markers of apoptotic cells, such as changes in the MtΔψ and MDA production.

It is apparent from the result that the MtΔψ level is increasing, which indicates the occurrence of hyperpolarization. Normally, intracellular mitochondria maintain a relative equilibrium in the MtΔψ at the endosomal membrane through the electron transport chain. However, when the cell is subjected to external stimuli, a large amount of H^+^ is transported from the mitochondrial matrix to the endosomal membrane space via the proton pump, leading to an abnormal increase in the MtΔψ [31]. Previous studies have also shown that a combination thymol and salicylic acid treatment induced the hyperpolarization of the MtΔψ in *Rhizopus stolonifera* [32]. Meanwhile, the increase in MDA content further confirmed our inference. Consequently, the antifungal mechanism of iturin A may be due to oxidative stress in which excess ROS induce the lipid peroxidation of cytomembranes and alterations in the MtΔψ which, in turn, induces fungal apoptosis.

Mitochondria are involved in a variety of cellular processes in fungi, including energy production, the transmission of information, and apoptosis [33]. Mitochondrial damage leads to dysfunctional metabolism and ROS accumulation. Our data demonstrated that mitochondrial dehydrogenase and ATPase levels were negatively correlated with iturin A concentrations. Y. et al. also reported that volatile organic compounds produced by *Pseudomonas fluorescens* ZX inhibited the activities of ATPase, MDH, and SDH in *Botrytis cinerea* species to a certain extent, subsequently impairing their respiratory metabolism [21]. In a proteomic analysis, many proteins associated with the TCA cycle, such as succinate dehydrogenase, citrate synthase, aconitate hydratase, and isocitrate dehydrogenase, were significantly downregulated in *A. niger* cells treated with iturin A. The result is consistent with previous findings and further demonstrates that iturin A is capable of disrupting the TCA cycle in *A. niger*.

Glycolysis/gluconeogenesis and the TCA cycle are the principal pathways of energy metabolism that provide energy supply to microorganisms [34]. According to the glycolysis pathway, the upregulation of hexokinase, glyceraldehyde 3-phosphate dehydrogenase (phosphorylating), and phosphoglycerate kinase in *A. niger* inhibited glycerol 3-phosphate production, leading to reduced pyruvate, which resulted in disordered carbohydrate transport [35]. Pyruvate, a product of glycolysis, is converted into acetyl–coenzyme A (acetyl CoA) by pyruvate decarboxylase. Acetyl CoA plays an important role in functional metabolism as an initiator of the TCA cycle. However, the downregulation of pyruvate decarboxylase decreased the progress of this reaction. A significant upregulation of aldehyde dehydrogenase and acetate increased the supply of acetyl CoA, which provided energy feedstock accumulation for the TCA cycle. In addition, the downregulation of aldehyde dehydrogenase decreased the rate of the microbially catalyzed ethanol conversion reaction and reduced the release of protons (H^+^), bringing about a microbial energy deficit. Thus, the disruption of mitochondrial energy metabolism pathways is associated with the mechanism of the inhibition of *A. niger*’s action by iturin A. To the best of our knowledge, such a mechanism has never been reported in the literature before.

## 5. Conclusions

In conclusion, the current findings indicate that iturin A effectively inhibits *A. niger*’s spore germination and mycelial growth. A biochemical analysis showed that the antifungal effect of iturin A was attributed to the disruption of cytomembrane integrity, oxidative stress, lipid peroxidation, and mitochondrial damage. In addition, through a proteomic analysis, we found that iturin A perturbed the metabolic pathways of *A. niger*, including the TCA cycle and glycolysis/gluconeogenesis, and led to the disruption of the mitochondrial energy metabolism, which provided the antifungal basis of iturin A at a molecular level. This study lays the foundation for further applications of iturin A as a natural antifungal agent to prevent *A. niger* contamination in the food industry.

## Figures and Tables

**Figure 1 jof-10-00172-f001:**
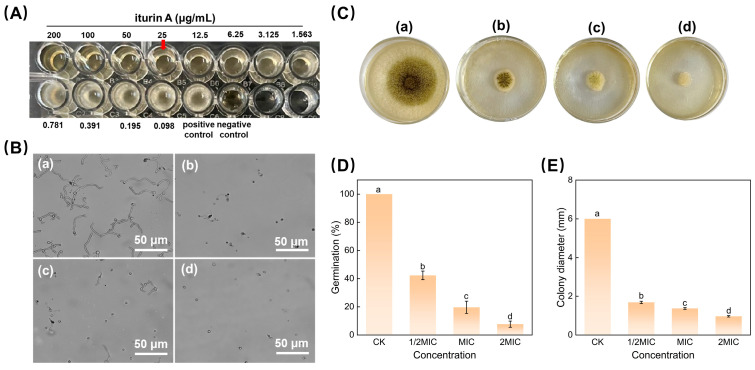
Iturin A inhibits the growth and development of *A. niger*. (**A**) Growth of *A. niger* on PDB medium containing a gradient increase in iturin A concentration from an initial concentration of 0.098–200 μg/mL. (**B**) Image of spores after 10 h of incubation in the dark at 28 °C. (**C**) Image of mycelial growth inhibition. (**D**) Statistical analysis of spore germination rate. (**E**) Statistical analysis of mycelial diameter. Images (**a**–**d**) were treated with 0, 1/2MIC, MIC, and 2MIC concentrations of iturin A, respectively. Bars indicate the SD of the mean (*n* = 3) values, and letters indicate statistically significant differences between groups at *p* < 0.05.

**Figure 2 jof-10-00172-f002:**
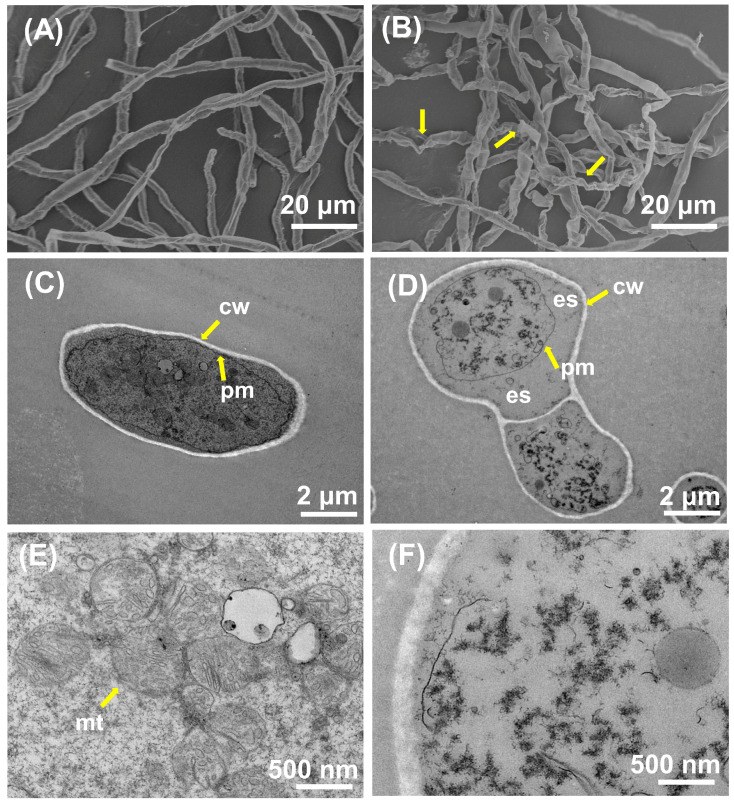
The SEM (**A**,**B**) and TEM (**C**,**D**) images of *A. niger* treated with different concentrations of iturin A. (**A**,**C**,**E**) Control and (**B**,**D**,**F**) treatment at the MIC. (**E**,**F**) are magnifications of images (**C**,**D**). (cw, cell wall; pm, plasma membrane; es, empty space; mt, mitochondria).

**Figure 3 jof-10-00172-f003:**
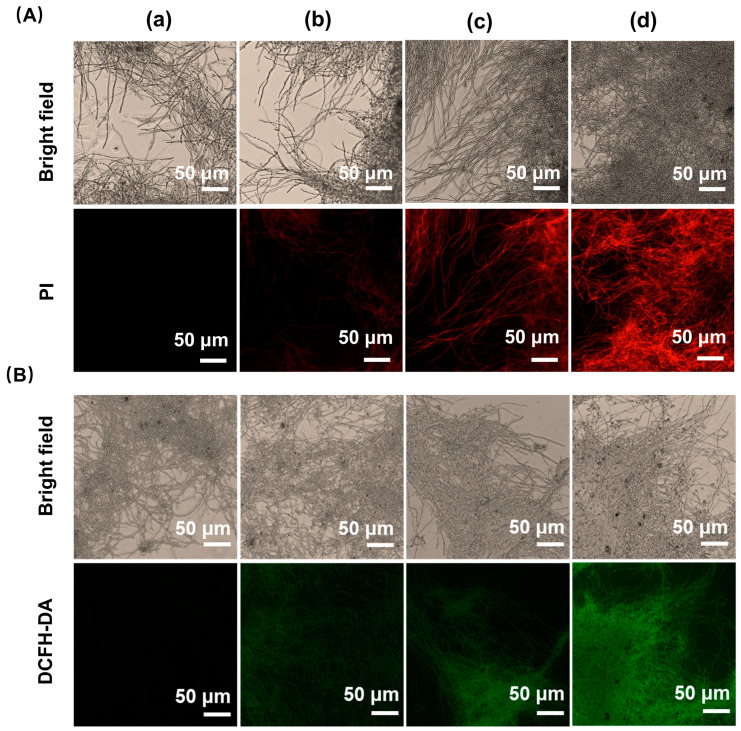
Effect of iturin A on the cytomembrane and reactive oxygen species (ROS) content of *A. niger*. (**A**) Effects of iturin A on the cytomembrane integrity of *A. niger*. (**B**) Effects of iturin A on ROS accumulation in *A. niger*. (**a**–**d**) *A. niger* was treated with 0, 1/2 MIC, MIC, and 2 MIC concentrations of iturin A, respectively.

**Figure 4 jof-10-00172-f004:**
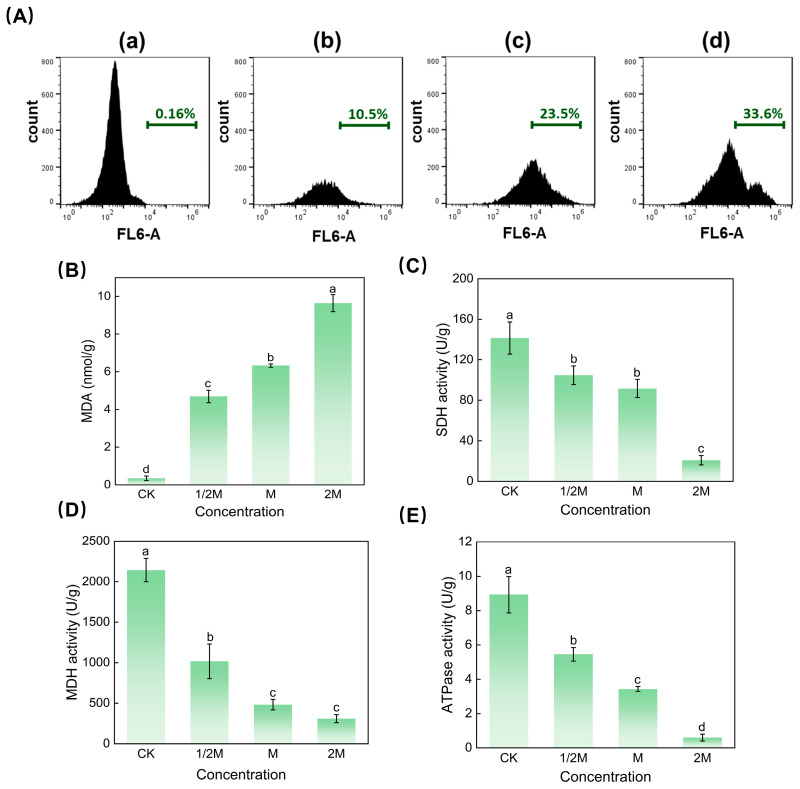
(**A**) Effect of iturin A on the MtΔψ of *A. niger*. (**a**–**d**) Represent *A. niger* treated with 0, 1/2 MIC, MIC, and 2 MIC concentrations of iturin A, respectively. (**B**) The content of MDA in *A. niger* after incubation at different concentrations of iturin A. (**C**) Efficacy of iturin A on the mitochondrial SDH activity of *A. niger*. (**D**) Efficacy of iturin A on the mitochondrial MDH activity of *A. niger*. (**E**) Efficacy of iturin A on the ATPase activity of *A. niger*. Bars indicate SD of the mean (*n* = 3) values, and letters indicate statistically significant differences between groups at *p* < 0.05.

**Figure 5 jof-10-00172-f005:**
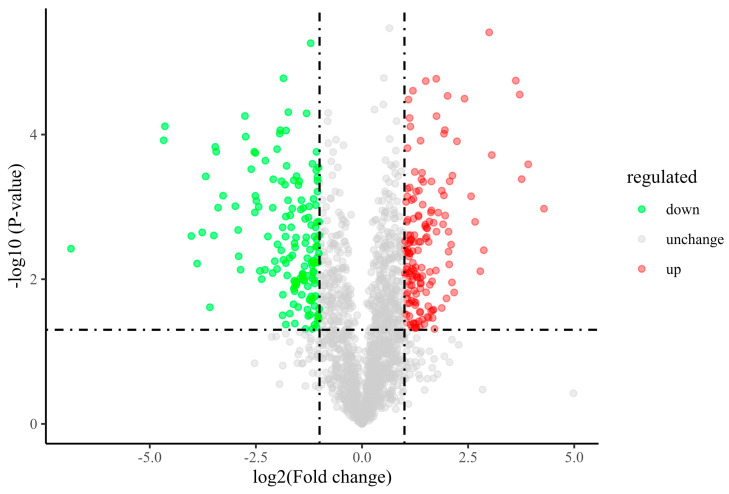
Changes of differentially expressed proteins in *A. niger* after iturin A treatment.

**Figure 6 jof-10-00172-f006:**
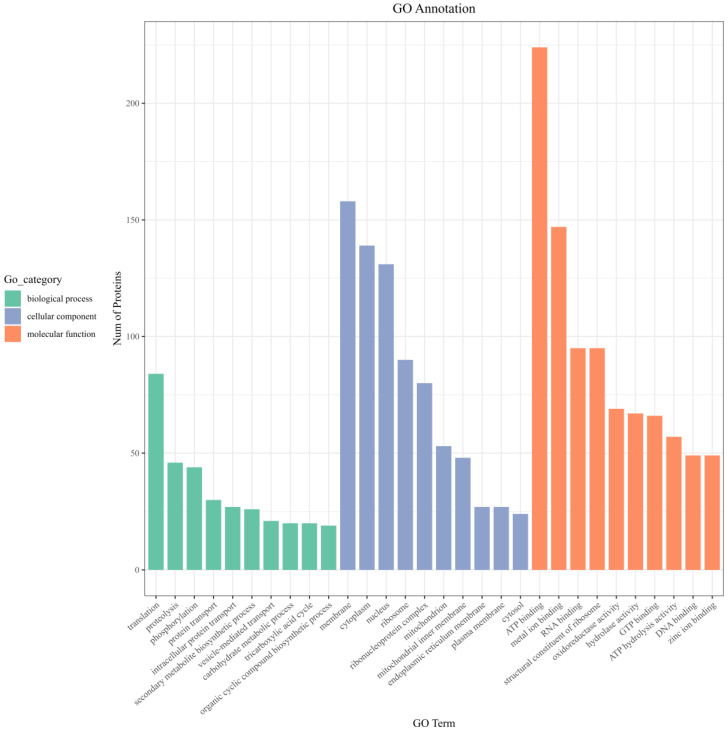
GO functional enrichment of differentially expressed proteins in *A. niger* after iturin A treatment.

**Figure 7 jof-10-00172-f007:**
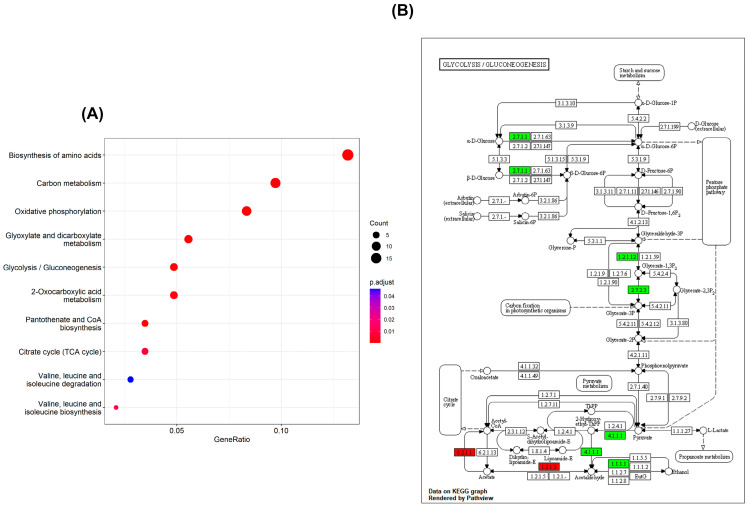
(**A**) KEGG functional enrichment of differentially expressed proteins in *A. niger* after iturin A treatment. (**B**) Changes in the glycolysis/gluconeogenesis pathways of *A. niger* after iturin A treatment. Red indicates differentially upregulated proteins; green indicates differentially downregulated proteins.

## Data Availability

Data are contained within the article.

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
