# Peer review of "Exploring Mechanisms of Antifungal Lipopeptide Iturin A from Bacillus against Aspergillus niger"

_jof, 2024, doi:10.3390/jof10030172_

Round 1

Reviewer 1 Report

In this publication, the authors aimed to study the mechanisms of iturin A's antifungal action on A. niger. Iturin A successfully prevents A. niger spore germination and mycelial development. They demonstrated using fluorescent probes that iturin A impaired cytomembrane integrity, resulting in oxidative stress, lipid peroxidation, and mitochondrial damage. Furthermore, by proteomic analysis, they demonstrated that iturin A altered the metabolic pathways of A. niger, including the TCA cycle, glycolysis / gluconeogenesis, and caused the disruption of mitochondrial energy production. Although the authors did not identify iturinA's molecular targets, the cellular processes they uncovered provided insight into potential mechanisms of action.

The experiments have been designed logically and the results are presented nicely.Only comment is

Authors used 2M and 2MIC at places which is confusing. They should Change 2M to MIC in the Fig 1D and 1E and line 234.

Reviewer 2 Report

It is known that mycoses are one of the most difficult diseases to treat, be they diseases of vertebrates, including humans, or plant diseases. In this regard, the search for microorganisms that produce antifungal metabolites is relevant. In this regard, analysis of the participation of bacterial metabolites of the Bacillus amyloliquefaciens strain BH072, especially non-pathogenic forms for humans, in the antifungal effect against the fungus Aspergillus niger ATCC16404. In this regard, the manuscript of the article by Wang et al., “Exploring Mechanisms of Antifungal Lipopeptide Iturin A from Bacillus Against Aspergillus niger,” is relevant. The authors of the manuscript carried out work to evaluate iturin A, previously isolated from the cultural filtrate of the bacterium B. amyloliquefaciens strain BH072, for antifungal activity against the fungus A. niger strain ATCC16404. The work convincingly proved that this lipopeptide is an important component responsible for the antifungal activity of metabolites of the bacterium B. amyloliquefaciens strain BH072, isolated from honey. The title of the work corresponds to the material presented. The conclusions are justified. The work is interesting, but during viewing, a number of amendments were discovered.

1. Summary. The bacterial strain must be specified (line 10). In the same way, you must specify the strain of the fungus (line 13).

2. Section 2.2. It is necessary to provide a link to the work where the procedure for isolating the bacterial strain B. amyloliquefaciens BH072 was described

3. Section 2.9. It is necessary to provide a link to the work where the method was described.

4. Section 2.10. It is necessary to provide a link to the work where the method was described.

5. Section 2.11. It is necessary to provide a link to the work or protocol of the developer's company, where the method was described.

6. Line 224. The name of the species is required in Italian font.

After minor technical corrections, the article can be published in the journal.

It is known that mycoses are one of the most difficult diseases to treat, be they diseases of vertebrates, including humans, or plant diseases. In this regard, the search for microorganisms that produce antifungal metabolites is relevant. In this regard, analysis of the participation of bacterial metabolites of the Bacillus amyloliquefaciens strain BH072, especially non-pathogenic forms for humans, in the antifungal effect against the fungus Aspergillus niger ATCC16404. In this regard, the manuscript of the article by Wang et al., “Exploring Mechanisms of Antifungal Lipopeptide Iturin A from Bacillus Against Aspergillus niger,” is relevant. The authors of the manuscript carried out work to evaluate iturin A, previously isolated from the cultural filtrate of the bacterium B. amyloliquefaciens strain BH072, for antifungal activity against the fungus A. niger strain ATCC16404. The work convincingly proved that this lipopeptide is an important component responsible for the antifungal activity of metabolites of the bacterium B. amyloliquefaciens strain BH072, isolated from honey. The title of the work corresponds to the material presented. The conclusions are justified. The work is interesting, but during viewing, a number of amendments were discovered.

1. Summary. The bacterial strain must be specified (line 10). In the same way, you must specify the strain of the fungus (line 13).

2. Section 2.2. It is necessary to provide a link to the work where the procedure for isolating the bacterial strain B. amyloliquefaciens BH072 was described

3. Section 2.9. It is necessary to provide a link to the work where the method was described.

4. Section 2.10. It is necessary to provide a link to the work where the method was described.

5. Section 2.11. It is necessary to provide a link to the work or protocol of the developer's company, where the method was described.

6. Line 224. The name of the species is required in Italian font.

After minor technical corrections, the article can be published in the journal.

Author Response

Reviewer #2:

We feel great thanks for your professional review work on our article. Your comments provided valuable insights to refine its contents and analysis. In this document, we try to address the issues raised as best as possible. And those changes are highlighted within the manuscript. All page numbers refer to the revised manuscript file with tracked changes.

  1. The bacterial strain must be specified (line 10). In the same way, you must specify the strain of the fungus (line 13).

Response: We have detailed labeling of bacterial and fungal strains in the abstract, which are Bacillus amyloliquefaciens (CGMCC No.8473) and Aspergillus niger (ATCC 16404), respectively (changes highlighted in red, Line 10). Hope the problem has been solved.

  1. Section 2.2. It is necessary to provide a link to the work where the procedure for isolating the bacterial strain B. amyloliquefaciens BH072 was described.

Response:

We have added references in Section 2.2 to work on the isolation and characterization of B. amyloliquefaciens BH072 (changes highlighted in red, Line 72). Hope the problem has been solved.

  1. Section 2.9. It is necessary to provide a link to the work where the method was described.

Response:

We have provided references to the determination of MDA content and added to the beginning of Section 2.9 (changes highlighted in red, Lines 168-169). Hope the problem has been solved.

  1. Section 2.10. It is necessary to provide a link to the work where the method was described.

Response:

We have provided references for the determination of ATPase and mitochondrial dehydrogenase and have added the relevant content at the beginning of Section 2.10 (changes highlighted in red, lines 176-177). Hope the problem has been solved.

  1. Section 2.11. It is necessary to provide a link to the work or protocol of the developer's company, where the method was described.

Response: We have contacted the developer's company to add specific methods for proteomics testing to the supplemental materials for reference. The supplementary materials are as follows. Hope the problem has been solved.

Spplementary materials

  1. Materials and Reagents

Table1 Main Materials and Reagents

Reagent name

Supplier

trypsin

Promega

acetonitrile

Fisher Chemical

trifluoroacetic acid

Sigma Aldrich

formic acid

Fluka

iodoacetamide

Sigma

dithiothreitol

Sigma

urea

Sigma

trichloroacetic acid

Sigma

protease inhibitor

Calbiochem

EDTA

Sigma

TEAB

Sigma

H2O

Fisher Chemical

Table2 Instrument list

Instrument

Model

Brand

Ultra High Performance Liquid Chromatograph

EASY-nLC 1200

Thermo Fisher Scientific

High Resolution Mass Spectrometry

Q-Exactive HF

Thermo Fisher Scientific

  1. Protein Extraction

Suitable protein extraction and precipitation purification methods were selected based on the samples, and the protein concentration of each sample was determined using the kit.

  1. Enzymatic Desalination

An appropriate amount of total protein was taken and added to the reducing agent buffer and reacted for 1 hour at 37°C. The reaction was carried out in the dark for 45 min at room temperature. Subsequently, IAA was added to a final concentration of 50 mM and the reaction was carried out in the dark for 45 min at room temperature. The reaction was quenched by the addition of 1M DTT solution of IAA, and the final concentration of urea was brought to less than 1 mol/L using 50 mmol/L NH4HCO3 solution. Finally, sequencing grade trypsin solution was added, incubated overnight at 37°C, and the digested peptides were desalted using a C18 solid phase extraction column.

  1. LC-MS Analysis

Mobile phase A was an aqueous solution containing 0.1% (v/v) formic acid; mobile phase B was acetonitrile containing 0.1% (v/v) formic acid. Peptides were dissolved in phase A of the liquid chromatography mobile phase and separated using the nanoACQUITY UPLC M-Class system (Waters, USA). The liquid gradient was set from 0 to 120 min, 8% to 100% mobile phase B. The flow rate was maintained at 300 nL/min.

The peptides were separated by the ultra-high performance liquid phase system and injected into the NSI ion source for ionization, and then analyzed by the Q Exactive HF mass spectrometer. The ion source voltage was set to 2.3 kV, and both the peptide precursor ions and their secondary fragments were detected and analyzed using a high-resolution Orbitrap in the Q Exactive HF. The scanning range of the primary mass spectrometer was set to 400-1800 m/z, the scanning resolution was set to 60000, and the secondary scanning resolution was set to 15000. The data acquisition mode uses a data-dependent (DDA) program, that is, after the first-level scan, the precursor ions of the top 20 peptides with the highest signal intensity are selected and sequentially entered into the HCD collision cell for fragmentation with a fragmentation energy of 28ev, and the second-level mass spectrometry is also performed sequentially. To improve the effective utilization of mass spectrometry, the automatic gain control (AGC) was set to 3E6, the signal threshold was set to 10000 ions, the maximum ion injection time was set to 50 ms, and the dynamic exclusion time of tandem mass spectrometry scanning was set to 45 s to avoid repeated scans of precursor ions.

Table3 LC mobile phase conditions

Time(min)

Flow rate(μL/min)

A%

B%

0

300

92

8

98

300

72

28

113

300

63

37

117

300

0

100

120

300

0

100

  1. Line 224. The name of the species is required in Italian font.

Response: We have revised the species name to Italian font (changes highlighted in red, Line 226). Hope the problem has been solved.

Thank you again for your positive comments and valuable suggestions to improve the quality of our manuscript, and we hope that the correction will meet with approval.